# Mulberry-Derived 1-Deoxynojirimycin Prevents Type 2 Diabetes Mellitus Progression via Modulation of Retinol-Binding Protein 4 and Haptoglobin

**DOI:** 10.3390/nu14214538

**Published:** 2022-10-28

**Authors:** Kamonpan Fongsodsri, Thanchanit Thaipitakwong, Kitiya Rujimongkon, Tapanee Kanjanapruthipong, Sumate Ampawong, Onrapak Reamtong, Pornanong Aramwit

**Affiliations:** 1Department of Tropical Pathology, Faculty of Tropical Medicine, Mahidol University, Ratchawithi Road, Ratchathewi, Bangkok 10400, Thailand; 2Center of Excellence in Bioactive Resources for Innovative Clinical Applications, Department of Pharmacy Practice, Faculty of Pharmaceutical Sciences, Chulalongkorn University, PhayaThai Road, Phatumwan, Bangkok 10330, Thailand; 3Department of Molecular Tropical Medicine and Genetic, Faculty of Tropical Medicine, Mahidol University, Ratchawithi Road, Ratchathewi, Bangkok 10400, Thailand; 4The Academy of Science, The Royal Society of Thailand, Dusit, Bangkok 10330, Thailand

**Keywords:** DNJ, insulin resistance, mulberry, proteomics, type 2 diabetes

## Abstract

Pre-diabetic or early-stage type 2 diabetes patients may develop an adverse diabetic progression, leading to several complications and increasing hospitalization rates. Mulberry leaves, which contain 1-deoxynojirimycin (DNJ), have been used as a complementary medicine for diabetes prevention and treatment. Our recent study demonstrated that mulberry leaf powder with 12 mg of DNJ improves postprandial hyperglycemia, fasting plasma glucose, and glycated hemoglobin. However, the detailed mechanisms are still unknown. This study investigates the effect of long-term (12-week) supplementation of mulberry leaves in obese people with prediabetes and patients with early-stage type 2 diabetes. Participants’ blood was collected before and after supplementation. The protein profile of the plasma was examined by proteomics. In addition, the mitochondrial function was evaluated by energetic and homeostatic markers using immunoelectron microscopy. The proteomics results showed that, from a total of 1291 proteins, 32 proteins were related to diabetes pathogenesis. Retinol-binding protein 4 and haptoglobin protein were downregulated, which are associated with insulin resistance and inflammation, respectively. For mitochondrial function, the haloacid dehalogenase-like hydrolase domain-containing protein 3 (HDHD-3) and dynamin-related protein 1 (Drp-1) displayed a significant increment in the after treatment group. In summary, administration of mulberry leaf powder extract in prediabetes and the early stage of diabetes can alleviate insulin resistance and inflammation and promote mitochondrial function in terms of energy production and fission.

## 1. Introduction

Type 2 diabetes mellitus (T2DM) is a chronic metabolic disorder and the most common found, making up over 90% of diabetes cases [1,2]. It is characterized by insulin resistance and the dysfunction of pancreatic islet β-cells, which leads to persistent hyperglycemia and altered lipid metabolism [3,4]. T2DM-related complications cause various life-threatening diseases, such as cardiovascular disease, cerebrovascular disease, neuropathy, nephropathy, retinopathy, foot disease, and peripheral gangrene [5,6,7]. In 2021, epidemiological data of the International Diabetes Federation (IDF) reported that 6.7 million people died due to diabetes and 537 million adults aged between 20 and 79 years had diabetes. In addition, the American Diabetes Association (ADA) reported that the economic burden associated with diabetes increased 60% from 2007 to 2017 [8]. The global medical expenditure on diabetes was evaluated at approximately USD 760 billion in 2019 [9]. Hence, preventing and alleviating the progression of diabetes is important to reduce the economic and health burden. Glycemic control is considered as the main therapeutic goal for the treatment of T2DM as it reduces the complications related to T2DM [10]. Furthermore, pharmacological treatment depends on comorbidities and patient-centered factors [11]. However, it has been reported that oral medication for diabetes treatment may cause adverse effects in some patients [12,13,14,15].

Currently, natural or herbal compounds have been shown to have antidiabetic activities [16,17]. Mulberry (*Morus alba* L.) leaves have been used in traditional Chinese medicines for several centuries [18] and have effects on diabetes therapy and its complications [19]. Interestingly, 1-deoxynojirimycin (DNJ), which is a well-known anti-diabetic agent because of its α-glucosidase inhibition activity, is the most-abundant phytochemical and bioactive compound in mulberry leaves [20,21,22]. Previous research regarding mulberry leaf extract suggested that postprandial glycemia was improved in individuals with impaired glucose metabolism in a dose-dependent manner [23]. The elevation of postprandial glucose and insulin secretion were significantly suppressed in healthy volunteers [24] and type 2 diabetic patients [25]. The plasma lipid profile in humans following the administration of mulberry leaf extract revealed that serum triglyceride levels decreased and lipoprotein levels were positively altered [26]. Nevertheless, the study of mulberry leaves on plasma protein expression is still limited.

Our previous randomized controlled clinical study of mulberry DNJ to find the optimal dose and examine the effect of the long-term administration of mulberry leaves revealed that 12 mg of mulberry DNJ was an optimal dose to reduce postprandial hyperglycemia, and there were no side effects [27]. Moreover, long-term ingestion of mulberry leaves showed a decrease in fasting plasma glucose (FPG). According to the results from a previous clinical study, the objective of this study aimed to examine the effect of long-term supplementation with mulberry leaves for the improvement of HbA1c in obese persons with prediabetes and patients with early-stage T2DM based on specific mechanisms. Plasma protein expression and mitochondrial functions were investigated using proteomics and electron microscopic studies. The significance of this study is that DNJ in mulberry leaf extract can be an upcoming organic candidate for inhibiting the progression of diabetes and improving clinical outcomes.

## 2. Materials and Methods

### 2.1. Mulberry Leaf Powder Preparation and Determination of DNJ

The mulberry leaf powder production process and DNJ determination were described previously [27]. Briefly, it was made from 50–70-day-old fresh mulberry leaves (Chiang Mai, Thailand), which were dried using electromagnetic radiation in a microwave oven. The dried mulberry leaves were finely ground and sieved through an 80-mesh sieve. Then, the powder was sterilized by gamma radiation and packed in sachets. Liquid chromatography with tandem mass spectrometry (API 3000 LC-MS/MS) (Sciex, Framingham, MA, USA) was applied to determine the DNJ in the mulberry leaf powder. The minimum DNJ in the mulberry leaf powder used in this study was 2.6 mg/g of dried weight.

### 2.2. Study Design

#### 2.2.1. Ethical Statement and Participant Inclusion

The study protocol was examined and approved by the Institutional Review Boards of the Royal Thai Army Medical Department, Phramonkutklao College of Medicine (No. Q038h/60). This study is a part of our previous clinical trial [27]. Recruitment criteria in our previous study were based on the ADA [11,28]. Briefly, participant recruitment involved obese persons aged between 20 and 65 years who had a BMI ≥ 25 kg/m^2^, a 75 g oral glucose tolerance test of 140–199 mg/dL, and an FPG of 100–140 mg/dL and/or 2 h postprandial glucose. The criteria for exclusion were as follows: persons who have been on antihyperglycemic drugs or had a record of diabetes treatment, high levels of aspartate transaminase and/or alanine transaminase ≥ three-times the upper limit of normal, diabetic complications and life-threatening conditions, creatinine ≥ 2.0 mg/dL, pregnancy or lactation, taking medication, supplements, or herbs affecting blood glucose levels within one month of study enrollment. Regarding our inclusion criteria, the overall population was 54 participants. They were randomly allocated into 2 groups: (i) 12 mg of DNJ as a treatment group (n = 28) and (ii) a non-treatment group (n = 26) over 12 weeks. According to the result from our previous trial, 12 mg of DNJ is an optimal dose to reduce postprandial hyperglycemia, improves HbA1C, and has no adverse effect. In addition, 12 participants in the treatment group had HbA1C reduction after 12-week supplementation of 12 mg of DNJ mulberry leaf powder. Thus, 12 participants from the treatment group were selected and used in the present study.

#### 2.2.2. Nutritional and Diet Control

Nutritional consulting with a licensed dietitian was performed on all participants. They were given the education about how to improve eating habits including a review and discussion of usual eating habits and developing an individualized eating pattern. Besides, participants were assigned to record a food diary such as foods, cooking methods of foods, snacks, drinks, and portions that they consumed on a representative day. Total caloric intake and the percentage of carbohydrate, protein, and fat consumed were calculated by the Thai Nutrisurvey^®^ software version 2.0 (developed by Faculty of Tropical Medicine, Mahidol university, Nakhon Pathom, Thailand). The assessment of the total caloric intake and macronutrients between the treatment and control groups at Weeks 0 and 12 is presented in Table 1. These determined that all participants had the same nutritional background before being subjected to the experiment.

#### 2.2.3. Human Experimental Protocol

Participants received 4.6 g of mulberry leaf powder along with nutritional control over the 12 weeks of the study. A single-ration sachet contained 4.6 g of mulberry leaf powder contained 12 mg of DNJ. For the instructions, the mulberry leaf powder from the sachet was mixed with 150 mL of warm water and ingested three times per day before meals.

#### 2.2.4. Specimen Collection and Plasma Preparation

Blood specimens of participants were collected at Week 0 and Week 12. The blood sample was centrifuged for blood fractionation, and the plasma was collected. According to the results of the previous study, 12 participants who had a reduction of HbA1c in the treatment group were selected for this study. The clinical characteristics of the participants are described in Table 2. These determined that all participants had the same clinical background before being subjected to the experiment. Their plasma specimens were used for proteomic analysis. Peripheral blood mononuclear cells (PBMCs) were collected from the buffy coat in a blood fractionation process for ultrastructure examination.

The plasma samples before treatment at Week 0 (n = 12) and after treatment at Week 12 (n = 12) were pooled within the same group, and each group was divided into three groups of four samples per group. Each of the pooled samples were considered as independent replicates and analyzed separately.

### 2.3. Label-Free Quantitative Proteomics of Plasma Samples from Mulberry Leaf Treatment

#### 2.3.1. Plasma Sample Extraction and Protein Size Separation

To investigate the effect of mulberry leaves on protein expression in the plasma of people with impaired glucose metabolism, such as obese persons with prediabetes and patients with early-stage T2DM, the plasma specimens were used to quantify the concentration of total protein using a Bradford protein assay (Bio-Rad^®^, Hercules, CA, USA). For the SDS polyacrylamide gel electrophoresis (SDS-PAGE), a 4% stacking and 12% separating gel was loaded with 30 µg of each crude protein from the plasma samples and run with a constant voltage at 120 V for 80 min in the gel electrophoresis unit (Bio-Rad^®^, USA), until the bromophenol blue dye reached the bottom of the gel.

#### 2.3.2. In-Gel Tryptic Digestion

The gel was stained using Coomassie Blue-R for 10 min and destained with destaining solution (30% methanol and 10% acetic acid) for 3 h. Afterwards, the destained gel was soaked in distilled water for 15 min before gel photography via a gel documentary system (Bio-Rad^®^, USA). Entire gel lanes were cut, and each lane was incised into 12 pieces, which were individually placed into 1.5 mL tubes. All gel pieces were dehydrated using 50% acetonitrile (Merck^®^, Kenilworth, NJ, USA) in HPLC-grade water (Merck^®^, Kenilworth, NJ, USA). They were incubated in 7 mM DTT in 50 mM ammonium bicarbonate (NH_4_HCO_3_) for 15 min at 60 °C. Proteins were alkylated in 250 mM iodoacetamide at room temperature in the dark for 30 min. The alkylation reaction was quenched by adding 7 mM DTT in 50 mM NH_4_HCO_3_. The gel pieces were dehydrated in 100% acetonitrile and dried at room temperature for 1 h. For trypsin digestion, trypsin in 50 mM NH_4_HCO_3_ was added to each gel piece at 37 °C for 16 h. Acetonitrile was added and incubated for 20 min at room temperature to collect the digested peptides. The samples were centrifuged at 10,000× *g* for 15 min at room temperature. The supernatant was collected and transferred into new 1.5 mL tubes. Then, the supernatant, which contained the digested peptides, was completely dried using a CentriVap Vacuum Concentrator (Labconco, Kansas City, MO, USA) at 40 °C.

#### 2.3.3. Protein Identification

The dried peptides were resuspended in 0.1% formic acid for LC-MS/MS analysis. They were subjected to the UltiMate^®^ 3000 Nano-LC system (Dionex, Altrincham, UK). The column was an Acclaim PepMap RSLC C18 75 μm × 15 cm (Thermo Scientific, Waltham, MA, USA) in stationary phase, at a flow rate of 300 nL/min. Mobile phase Solutions A and B consisted of 0.1% formic acid and 80% acetonitrile in 0.1% formic acid, respectively. The initial mobile phase step was performed using 4% of Solution B for 5 min. Gradient conditions were used to elute the peptides, from 4% to 50% of Solution B for 30 min, holding for 5 min, and then, returning to the initial phase for 10 min. Subsequently, the eluted peptides were subjected to a positive electrospray ionization system coupled with a microTOF-Q II (Bruker, Bremen, Germany). The MS and MS/MS spectra covered the mass range of *m*/*z* 400–2000 and *m*/*z* 50–1500, respectively. The raw data files were converted to mascot generic files using the DataAnalysis software version 3.4 (Bruker, Bremen, Germany) Mascot Daemon version 2.3.02 (Matrix Science, London, UK) was used for the identification and quantification of the proteins. The protein database was obtained from SwissProt specific to *Homo sapiens* (searching date: 2 November 2020). For the search parameter settings, the methionine oxidation and carbamidomethylation of cysteines were set as the fixed modification and variable modification, respectively. The identified proteins with a significant score (*p* < 0.05) that appeared in at least two samples in the group were reported. The semi-quantitative protein expression from peptide counts was determined by the Exponentially Modified Protein Abundance Index (emPAI) [29]. The change of protein expression was analyzed by comparing the emPAI count before and after treatment. The characterization of biological processes and proteins associated with organs was accomplished using the UniProt database (www.uniprot.org (accessed on 30 December 2020)). The STRING software was used to predict protein–protein interaction networks (https://sring-db.org/ (accessed on 30 December 2020)).

### 2.4. Extraction of Mitochondria from Peripheral Blood Mononuclear Cells before and after Mulberry Leaf Treatment

Mitochondria were extracted form PBMCs using a Mitochondria Isolation Kit (My BioSource, San Diego, CA, USA). In brief, mitochondria isolation buffer was added to the sample and vortexed for 5 s, followed by incubation on ice for 2 min. Then, the sample had added 10 µL of Reagent A and was vortexed for 5 s before incubation on ice for 5 min, vortexing every minute for 5 s. The sample was then centrifuged at 600× *g* for 10 min at 4 °C. The supernatant was transferred into a separate tube and centrifuged at 7000× *g* for 10 min at 4 °C. The supernatant was discarded, and the pellet was washed with mitochondria isolation buffer. Afterwards, the sample was repeatedly centrifuged, and the pellet was resuspended in storage buffer.

### 2.5. Mitochondrial Immunogold Labeling from PBMCs

To examine the specific markers that relate to PBMCs mitochondrial functions before and after mulberry leaf treatment, immunogold labeling was performed. The extracted mitochondria were pooled in each group, before and after treatment. They were fixed in 2.5% glutaraldehyde for 1 h and 1% osmium tetroxide for 1 h, respectively. They were dehydrated in serial-graded ethanol and infiltrated with a series of LR white resin (EMS^®^, Houston, TX, USA). The samples were embedded in pure LR white (EMS^®^, USA) and polymerized at 60 °C for 48 h. The mitochondrial samples were cut into 100 nm. Subsequently, they were incubated with primary antibodies, including the fusion marker antioptic atrophy 1 (OPA-1), the fission marker antidynamin-related protein 1 (Drp-1), the energy marker antihaloacid dehalogenase-like hydrolase-domain-containing protein 3 (HDHD-3), and nuclear factor erythroid-2-related factor 2 (Nrf-2) for 1 h and washed using 0.1% BSA in PBS. Then, goat anti-rabbit or mouse IgG conjugated with ultra-small (3–5 nm) or 10 nm gold particles (EMS^®^, USA) were applied to the sections and washed several times using 0.1% BSA in PBS. A silver enhancement (Aurion R-Gent SE-EM kit, EMS, USA) was used to improve the contrast of gold particle labelling after washing the mitochondrial sections with distilled water. They were stained with lead citrate and uranyl acetate prior to observation under a transmission electron microscope. The number of labelled gold particles was counted as immunolabeling per field.

### 2.6. Statistical Analysis

Statistical analysis was performed using GraphPad Prism^®^ version 9 (GraphPad, San Diego, CA, USA). The immunogold labelling was analyzed using the paired *t*-test. The difference between the before and after treatment groups is presented as the mean ± the standard deviation (SD) and the range. The statistical significance was defined at *p*-values < 0.05.

## 3. Results

### 3.1. Identification of the Protein Profile in Plasma before and after Mulberry Leaf Treatment

To determine the effects of mulberry leaves and specific mechanisms that involve reduced the HbA1C in the plasma of people with impaired glucose metabolism, such as obese persons with prediabetes and patients with early-stage T2DM, label-free proteomics was performed to discover the plasma protein profile. The results were acquired from plasma specimens of the selected cases before (Week 0) and after (Week 12) mulberry leaf treatment. The crude protein of the plasma samples was extracted, and extracted proteins (30 µg) were separated using SDS-PAGE gels (Figure 1A). Each gel lane was excised into 12 pieces and individually digested in a tryptic digestion process prior to mass spectrometry. A total of 1291 proteins were identified from the plasma before and after mulberry leaf treatment. The protein expression in the before and after treatment groups found that 308 and 204 different proteins were presented only in the before and after treatment groups, respectively (Appendix A), while 779 proteins were expressed in both groups (Figure 1B). The overlapping 779 proteins were analyzed, and the result showed 7 up-expressed and 4 down-expressed proteins (Appendix A).

### 3.2. Differentiation of Proteins, Their Function, and Associated Pathways

All of the proteins from the 308 proteins in the before treatment group, the 204 proteins in the after treatment group, and the 7 up-expressed and 4 down-expressed proteins in both groups were identified according to biological functions. Gene ontology enrichment analysis was performed to focus on the specific proteins related to metabolic disturbance in T2DM pathogenesis. The result revealed 20 unique proteins presented in the before treatment group, 10 proteins found in the after treatment group, and 2 down-expressed proteins in both groups. These specific proteins were closely associated with the pathogenesis and pathophysiology of metabolic disturbance in T2DM. The functional protein interaction and associated pathway of the different proteins were explored using the STRING database and KEGG pathway database, respectively. The associated pathways were then classified into potential 13 pathways and categorized into 4 major groups: (i) metabolic regulation (21.62%), (ii) inflammatory response (24.32%), (iii) immune response (27.03%), and (iv) ECM constituents and organization (27.03%) (Figure 2 and Figure 3). All differential proteins are illustrated in Table 3.

### 3.3. Immunogold Labeling of PBMCs’ Mitochondria from before and after Treatment

To investigate the effect of mulberry leaf treatment in term of PBMCs’ mitochondrial functions, immunogold labeling was performed. The specific markers related to mitochondrial functions were examined, including the fusion marker (OPA-1), fission marker (Drp-1), and energy production markers (HDHD-3 and Nrf-2). The result demonstrated that the expression of HDHD-3 (Figure 4A–C) and Drp-1 (Figure 4D–F) in the PBMCs’ mitochondria after treatment was significantly higher than in the PBMCs’ mitochondria before treatment. However, the expression of Nrf-2 (Figure 4G–I) and OPA-1 (Figure 4J–L) was not significantly different between the PBMCs’ mitochondria in the before and after treatment groups.

## 4. Discussion

The present study investigated the effect of long-term mulberry leaf supplementation and specific mechanisms that improved the HbA1C in the plasma protein profile and mitochondrial functions in patients with early-stage T2DM and obese persons with prediabetes. The results of our experiment demonstrated that a total of 1291 plasma proteins were identified from the before and after treatment groups. We found 308 unique proteins present in the before treatment plasma, 204 unique proteins in the after treatment group, and 11 unique proteins in both groups (Figure 1). These proteins were then classified to determine the biological functions and specific proteins that relate to T2DM pathogenesis, identifying 20 proteins in the before treatment group, 10 proteins in the after treatment group, and 2 down-expressed proteins present in both groups (Table 3). These 32 proteins were identified with four associated pathways: (i) metabolic regulation, (ii) inflammatory response, (iii) immune response, and (iv) ECM constituents and organization (Figure 2). Moreover, the percentage of the 32 different proteins involved in the four pathways was 21.62% for the metabolic regulation pathway, 24.32% for the inflammatory response pathway, 27.03% for the immune response pathway, and 27.03% for the ECM constituents and organization pathway, respectively (Figure 3). Interestingly, proteins related to insulin resistance and inflammation-induced hyperglycemia (retinol-binding protein 4 and haptoglobin, respectively) were downregulated after treatment. In addition, it has been reported that mitochondrial dysfunction related to T2DM causes the lowering of ATP production and β-oxidation and a higher production of reactive oxygen species, leading to insulin resistance [30,31,32,33,34]. Therefore, to determine the mitochondrial functions of the PBMCs in terms of homeostatic proteins (fusion and fission) and energy production, immunogold labeling was performed in this study. The results showed the significant upregulation of HDHD-3 and Drp-1 in the after treatment group (Figure 4). According to these results, mulberry leaf powder extract treatment contributes to an increase in the energy maintenance protein and mitochondrial fission, which facilitates the removal of damaged mitochondria [35].

In the present study, protein expressions in the after mulberry leaf treatment group were discovered and categorized into metabolic regulation, ECM constituent and organization, and immune response pathways. Insulin receptor substrate 2 (IRS2) was found, which is a major insulin receptor substrate in the insulin signaling pathway and has been indicated to play a crucial role in glucose metabolism and adipose tissue [36]. The upregulated expression of IRS2 can prevent the progression of diabetes [37]. Oxysterol LXR-alpha receptor expression serves as an endogenous ligand in cholesterol metabolism and stimulates excess cholesterol flux via the reverse cholesterol transport by high-density lipoprotein [38]. This result indicated that long-term mulberry leaf supplementation containing 12 mg of DNJ improves the metabolic pathway by altering glucose and cholesterol metabolisms via the expression of IRS2 and oxysterol LXR-alpha proteins. Interestingly, we found the new proteins associated with immune response after mulberry leaf ingestion were pyrin and NACHT-, LRR-, and PYD-domain-containing protein 7. According to this finding, mulberry leaf supplementation may improve the immunological protein level. Nevertheless, the evidence and the function of these two proteins in relation to diabetes are still unclear. Furthermore, integrin β-6 was present in the after treatment group, which is a transmembrane receptor that mediates cell adhesion, migration, and proliferation. Jacobsen et al. suggested that integrin β-6 may prohibit re-epithelialization of diabetic wounds [39]. Dynamin 1-like protein (drp1) was also found in this group; its function is to regulate mitochondrial fission [40]. It was reported that metformin can suppress excessive mitochondrial fission by inhibiting drp1 protein activity and restoring mitochondrial homeostasis in diabetic mice [41]. Son of sevenless protein 1 (SOS1) is involved in the tyrosine phosphorylation of insulin receptor substrate 1 (IRS1) in the insulin signaling pathway [42]. CREB-binding protein acts as a transcriptional factor and is located in the nucleus. Depletion of CREB-binding protein in the vasculature may enhance atherosclerosis in a diabetes in vivo model [43]. Finally, the proteins in this group associated with metabolic regulation such as the insulin signaling pathway and cholesterol metabolism may enhance the metabolic pathway, leading to an improvement in insulin resistance and prohibit the progression of diabetes. Pyrin and NACHT-, LRR-, and PYD-domain-containing protein 7 were found in our study; these two proteins are involved in the immune response, and their function in diabetes is still ambiguous.

One protein discovered in both the before and after treatment groups, retinol-binding protein 4 (RBP4), was downregulated in our study. RBP4 is a protein secreted by adipocytes and is a useful biomarker for systemic insulin resistance [44]. Additionally, the increase in serum RBP4 levels correlated with insulin resistance among the subjects with T2DM, obesity, and impaired glucose tolerance [45,46]. The other downregulated protein was a haptoglobin (Hp), which is an acute-phase protein in inflammation. Elevated Hp is also influenced by inflammation, obesity, hypertension, and their polymorphism [47]. In agreement with this finding, this information could suggest that mulberry leaves reduce insulin resistance and inflammation by the downregulation of RBP4 and Hp proteins, respectively.

Before mulberry leaf treatment, the discovered proteins were mostly related to the immune, inflammatory response, and ECM constituent and organization pathways and found to be only slightly related to the metabolic regulation pathway (Table 3). The proteins in this group were mainly found to be various types of collagen, such as type 2, type 4, and type 6 collagen. However, we found a few proteins present in the ECM constituent and organization pathway related to T2DM in the after treatment group. Moreover, other proteins were predominantly present in the immune and inflammatory response pathways, such as proteins involving the activation of NF-κB, the acute phase reactant, and the TNF-α receptor (Table 3). The proteins shown in the metabolic regulation pathway included phosphatidylinositol 3-kinase regulatory subunit beta (PIK3R2) and the insulin receptor, an important enzyme and receptor in the insulin signaling pathway [48], and putative hexokinase HKDC1, which is a hexokinase involved in glucose metabolism [49]. Although, we found some proteins that may play an important role in the insulin signaling pathway, the DNJ in the mulberry leaves could not maintain the prolonged action of those proteins after mulberry leaf supplementation. On the other hand, DNJ improved insulin signaling and insulin resistance through the expression of these proteins in the after treatment group.

The limitations of this study should be noted, regarding our limited sub-population of participants, who were selected from our previous trial using the reduction of HbA1C criterion [27]. However, our results provide preliminary data for further studies associated with the effect of DNJ on insulin resistance and inflammation.

## 5. Conclusions

The effect of the long-term supplementation of mulberry leaf powder and specific mechanisms that involve reducing the HbA1C in individuals with prediabetes and early-stage T2DM was examined using plasma via proteomic and mitochondrial immunogold labeling. The plasma protein profile demonstrated that 12 mg of DNJ in mulberry leaf powder could enhance metabolic regulation by modulating the expression of signaling proteins in the insulin signaling pathway and an endogenous ligand in cholesterol metabolism. The downregulation of secreted adipocyte and acute-phase proteins may reduce insulin resistance and inflammation. Mitochondrial energy homeostasis and fission proteins increased after mulberry leaf supplementation. Our study provides information regarding the effect of long-term mulberry leaf supplementation and specific mechanisms, which could improve clinical outcomes and be further developed as an alternative supplementation for diabetes patients.

## Figures and Tables

**Figure 1 nutrients-14-04538-f001:**
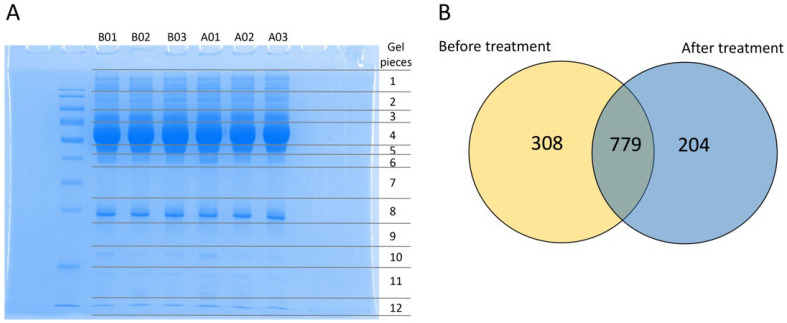
The label-free quantitative proteomic analysis of plasma specimens from the mulberry leaf treatment study. (**A**) SDS-PAGE gel stained with Coomassie Blue-R of plasma proteins from the before and after treatment group. Each lane represents the plasma samples from before treatment with triplication (B01, B02, and B03, respectively) and after treatment with triplication (A01, A02, and A03, respectively). (**B**) The illustration of the total of 1291 proteins revealed 308 and 204 differentially expressed proteins expressed in the before and after treatment groups, respectively. The overlapping 779 proteins indicated the number of altered proteins shared between the before and after treatment groups.

**Figure 2 nutrients-14-04538-f002:**
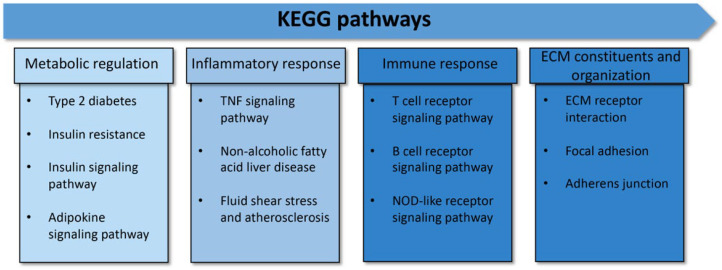
Protein categorization by KEGG pathways. The 13 associated pathways in metabolic disturbance in type 2 diabetes were categorized into 4 major groups: 1. metabolic regulation, 2. inflammatory response, 3. immune response, and 4. ECM constituents and organization.

**Figure 3 nutrients-14-04538-f003:**
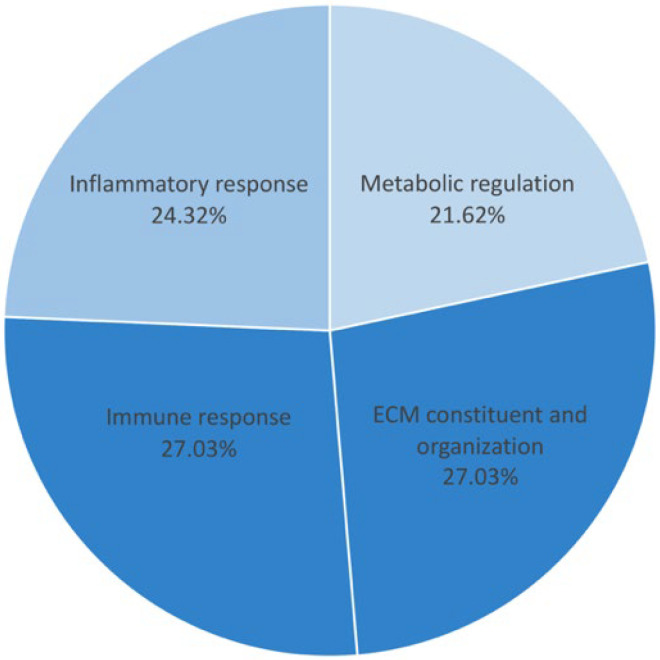
The percentage of proteins in 4 major pathways. The metabolic regulation pathway was 21.62%; the inflammatory response pathway was 24.32%; the immune response and ECM constituents and organization pathway were 27.03%.

**Figure 4 nutrients-14-04538-f004:**
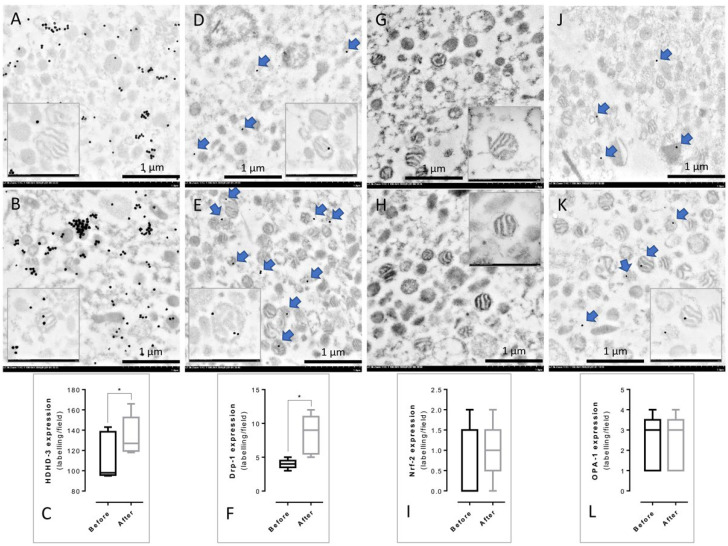
Immunogold labeling of the PBMCs’ mitochondria from the before and after treatment groups. Before treatment (**A**,**D**,**G**,**J**) and after treatment (**B**,**E**,**H**,**K**) with gold labeling (arrow) were revealed. The presence of mitochondrial function markers was represented in HDHD-3 (**A**–**C**), Drp-1 (**D**–**F**), Nrf-2 (**G**–**I**), and OPA-1 (**J**–**L**). All data in the bar graph are presented as the mean ± SD, * *p* < 0.05.

**Table 1 nutrients-14-04538-t001:** The mean total calorie and macronutrient intake per day at Weeks 0 and 12 (n = 54).

	Timepoint of Measurement	Treatment Group (n = 28)	^#^*p*-Values	Control Group (n = 26)	^#^*p*-Values	* *p*-Values
Total calories (kcal/day)	Week 0	1707.75 ± 144.16 (1561.6–2001.8)	0.409	1648.29 ± 173.75 (1535.7–1920.9)	0.156	0.083
	Week 12	1777.41 ± 141.94 (1479.7–1804.4)		1626.62 ± 152.42 (1458.0–1780.2)		
Carbohydrate (g/day)	Week 0	169.13 ± 56.71 (88.1–215.4)	0.696	163.44 ± 69.08 (123.7–197.8)	0.534	0.482
	Week 12	165.94 ± 52.52 (90.4–189.0)		156.78 ± 41.39 (110.6–192.3)		
Fat (g/day)	Week 0	52.33 ± 14.10 (33.8–80.2)	0.224	57.29 ± 16.27 (39.9–101.5)	0.732	0.152
	Week 12	55.98 ± 9.27 (30.5–90.9)		56.13 ± 8.54 (48.1–104.2)		
Protein (g/day)	Week 0	64.16 ± 17.66 (47.2–88.6)	0.325	63.86 ± 14.40 (51.8–96.6)	0.077	0.748
	Week 12	70.39 ± 12.19 (50.1–90.1)		68.34 ± 9.17 (45.6–111.4)		

Data are reported as the mean ± S.D. with the range (min–max). ^#^
*p*-values for within-group comparison (vs. Week 0) analyzed by a paired *t*-test. * *p*-values for between-group comparison (vs. control) analyzed by one-way ANOVA.

**Table 2 nutrients-14-04538-t002:** Clinical characteristic of the participants.

	Time Points of Measurement	Treatment Group (n = 28)	^#^ *p*	Control Group (n = 26)	^#^ *p*	* *p*
Weight (kg)	Week 0	78.06 ± 13.98		81.31 ± 20.79		
	Week 12	77.43 ± 14.30	0.075	80.92 ± 14.06	0.098	0.352
BMI (kg/m^2^)	Week 0	30.06 ± 4.06		31.01 ± 5.85		
	Week 12	29.80 ± 4.07	0.075	30.82 ± 3.86	0.094	0.272
Waist circumference (cm)	Week 0	100.52 ± 11.21		102.92 ± 15.67		
	Week 12	100.77 ± 11.32	0.586	101.98 ± 11.08	0.459	0.693
PPG-30 (mg/dL)	Week 0	173.89 ± 17.52		172.04 ± 21.48		
	Week 12	163.14 ± 20.63	0.018	166.96 ± 19.78	0.210	0.491
PPG-60 (mg/dL)	Week 0	186.61 ± 29.95		191.46 ± 31.96		
	Week 12	182.14 ± 24.81	0.528	190.65 ± 34.30	0.865	0.358
PPG-90 (mg/dL)	Week 0	171.21 ± 33.19		177.46 ± 40.25		
	Week 12	169.29 ± 29.92	0.734	174.04 ± 45.20	0.482	0.648
PPG-120 (mg/dL)	Week 0	141.14 ± 32.97		155.12 ± 37.53		
	Week 12	163.14 ± 20.63	0.251	166.96 ± 19.78	0.626	0.173
FPI (µIU/mL)	Week 0	12.55 ± 7.85		14.39 ± 6.02		
	Week 12	12.41 ± 6.66	0.188	14.05 ± 7.32	0.643	0.399
HOMA-IR	Week 0	3.60 ± 2.18		3.84 ± 1.72		
	Week 12	3.18 ± 1.78	0.057	3.78 ± 2.18	0.781	0.278
TC (mg/dL)	Week 0	194.99 ± 35.53		200.66 ± 37.93		
	Week 12	181.58 ± 26.64	0.002	199.49 ± 33.76	0.786	0.034
TG (mg/dL)	Week 0	131.63 ± 45.22		145.03 ± 59.99		
	Week 12	123.52 ± 35.87	0.148	138.10 ± 51.38	0.583	0.193
HDL-C (mg/dL)	Week 0	54.89 ± 14.35		53.56 ± 9.82		
	Week 12	52.48 ± 10.83	0.098	54.63 ± 8.66	0.302	0.424
LDL-C (mg/dL)	Week 0	139.06 ± 33.80		137.46 ± 34.25		
	Week 12	131.66 ± 29.54	0.090	137.25 ± 29.81	0.959	0.492
A1C (%)	Week 0	5.8 ± 0.4		5.7 ± 0.3		
	Week 12	5.7 ± 0.3	0.011	5.8 ± 0.4	0.100	0.551
Creatinine (mg/dL)	Week 0	0.81 ± 0.17		0.72 ± 0.18		
	Week 12	0.76 ± 0.22	0.149	0.70 ± 0.17	0.191	0.244
eGFR	Week 0	92.79 ± 14.86		99.75 ± 17.39		
	Week 12	88.33 ± 24.32	0.255	100.75 ± 16.48	0.467	0.034
AST (U/L)	Week 0	21.92 ± 4.84		19.39 ± 5.19		
	Week 12	21.83 ± 6.17	0.927	22.27 ± 8.19	0.113	0.825
ALT (U/L)	Week 0	23.21 ± 7.78		19.24 ± 7.05		
	Week 12	25.24 ± 13.09	0.345	23.57 ± 17.49	0.114	0.692

Data are reported as the mean ± S.D. * *p*-values for between-group comparison (vs. control) analyzed by one-way ANOVA; ^#^
*p*-values for within-group comparison (vs. Week 0) analyzed by a paired *t*-test. kg: kilogram; BMI: body mass index; kg/m^2^: kilogram per square meter; FPG: fasting plasma glucose; mg/dL: milligram per deciliter; PPG: postprandial plasma glucose; A1C: glycated hemoglobin; FPI: fasting plasma insulin; µIU/mL: micro international units per millimeter; HOMA-IR: homeostatic model assessment of insulin resistance; TC: total cholesterol; TG: triglyceride; HDL-C: high-density lipoprotein cholesterol; LDL-C: low-density lipoprotein cholesterol; eGFR: estimated glomerular filtration rate; AST: aspartate transaminase; ALT: alanine aminotransferase; U/L: units per liter.

**Table 3 nutrients-14-04538-t003:** The list of differential proteins associated with metabolic disturbance in type 2 diabetes.

Groups	Accession No.	Gene	Protein Name	Protein Function	Pathways	Score	Coverage (%)
Before Treatment	P85B_HUMAN	*PIK3R2*	Phosphatidylinositol 3-kinase regulatory subunit beta	Regulation of glucose metabolism	1	43	8.1
	INSR_HUMAN	*INSR*	Insulin receptor	Receptor of insulin		49	8.1
	HKDC1_HUMAN	*HKDC1*	Putative hexokinase HKDC1	Intermediate of glucose uptake in peripheral tissues		38	16.5
	TNR1A_HUMAN	*TNFRSF1A*	Tumor necrosis factor receptor superfamily member 1A	Receptor of TNF-α	2	41	16.7
	MP2K6_HUMAN	*MAP2K6*	Dual-specificity mitogen-activated protein kinase kinase 6	Mediator of MAPK and JNK activation		36	15.9
	ANGT_HUMAN	*AGT*	Angiotensinogen			131	10.5
	A1AG2_HUMAN	*ORM2*	Alpha-1-acid glycoprotein 2	Positive acute phase reactant		59	22.9
	FETUA_HUMAN	*AHSG*	Alpha-2-HS-glycoprotein			55	15.3
	IKKA_HUMAN	*CHUK*	Inhibitor of nuclear factor kappa-B kinase subunit alpha	Mediator of NF-kB activation	2 and 3	42	11.7
	NEMO_HUMAN	*IKBKG*	NF-kappa-B essential modulator			36	11
Before treatment	NOD2_HUMAN	*NOD2*	Nucleotide-binding oligomerization domain containing protein 2	Immune response		41	5.8
	FCN3_HUMAN	*FCN3*	Ficolin-3	Immune response	3	41	7
	TEC_HUMAN	*TEC*	Tyrosine-protein kinase Tec			66	19.2
	CO2A1_HUMAN	*COL2A1*	Collagen alpha 1(II) chain	Structural and ECM constituents	4	34	9.8
	CO4A3_HUMAN	*COL4A3*	Collagen alpha 3(IV) chain			53	9.6
	CO6A3_HUMAN	*COL6A3*	Collagen alpha 3(VI) chain			75	4.8
	TRI46_HUMAN	*TRIM46*	Tripartite motif-containing protein 46	Cell interaction and communication		49	19.1
	EMIL2_HUMAN	*EMILIN2*	EMILIN-2			35	5.3
	ATS12_HUMAN	*ADAMTS12*	A disintegrin and metalloproteinase with thrombospondin motifs 12			54	7.0
	ZO1_HUMAN	*TJP1*	Tight junction protein ZO-1			49	5.3
After treatment	IRS2_HUMAN	*IRS2*	Insulin receptor substrate 2	Mediator of insulin action	1	54	6.7
	NR1H3_HUMAN	*NR1H3*	Oxysterols receptor LXR-alpha	Nuclear receptor in regulation of lipid metabolism		35	23
	SOS1_HUMAN	*SOS1*	Son of sevenless homolog 1	Signaling pathway (G-protein-coupled receptor)	1 and 3	47	6.3
	SOS2_HUMAN	*SOS2*	Son of sevenless homolog 2			38	8.6
	MEFV_HUMAN	*MEFV*	Pyrin	Inflammasome and inflammatory response	3	49	8.3
	NALP7_HUMAN	*NLRP7*	NACHT, LRR and PYD domains containing protein 7			40	6.3
	DNM1L_HUMAN	*DNM1L*	Dynamin-1-like protein	Cellular process		44	12.2
	COMP_HUMAN	*COMP*	Cartilage oligomeric matrix protein	Cellular process	2	46	13.7
	ITB6_HUMAN	*ITGB6*	Integrin beta 6	Cell interaction and communication		41	15.2
	CBP_HUMAN	*CREBBP*	CREB-binding protein	Signaling pathway (Notch)		51	5.6
Down- regulation	RET4_HUMAN	*RBP4*	Retinol-binding protein 4	Adipokine	1	133	24.9
	HPT_HUMAN	*HP*	Haptoglobin	Positive acute phase reactant	2	1260	53.7

Pathways in this Table 2 as follows: 1—metabolic regulation, 2—inflammatory response, 3—immune response, and 4—ECM constituents and organization.

## Data Availability

Not applicable.

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
