# Peer review of "Mulberry-Derived 1-Deoxynojirimycin Prevents Type 2 Diabetes Mellitus Progression via Modulation of Retinol-Binding Protein 4 and Haptoglobin"

_nutrients, 2022, doi:10.3390/nu14214538_

Round 1

Reviewer 1 Report

The findings are interesting but without data for the control group, the entire results are not reliable because we do not know if the results were due to chance or a true effect of the extract. I understand that the authors tried to present this as a limitation of the study but I suggest that they consider adding the control group data for all experiments and discussing the same for better clarity and reliability.

 Other concerns are:

1. Title Is too wordy. I will suggest something like: “Mulberry-derived 1-Deoxynojirimycin (DNJ) Prevents 2 Diabetes Mellitus Progression via modulation of Retinol-binding Protein 4 and Haptoglobin” or something short and concise.

2. There are many grammatical errors that make the manuscript difficult to follow. Proofread to improve clarity.

3. The abstract should be revised to improve clarity. There are a lot of unnecessary details that can be cut from the abstract from Line 17 to Line 26. Also, the sentence starting from Line 26 to Line 31 does not summarize the results/discussion properly.

4. Line 101-108 should be revised. It is confusing that an experiment from a previous study was presented as if it was part of the present study. This should be revised to show that in the previous study with 4 groups, one (12mg DNJ) was considered optimal and was used in the present study.

5. Line 230-234 is confusing. Please revise to indicate that 779 were present in both groups, while 308 and 204 were found only in the before and after treatment groups respectively.

Reviewer 2 Report

Major Revision:

Kamonpan and colleagues' paper aims to demonstrate the beneficial effects of mulberry leaves in pre-diabetic or T2DM patients. Through their analyzes, they have shown a positive impact on the improvement of factors related to insulin resistance and the improvement of mitochondrial well-being.

However, there is a significant limitation to the study regarding diet.

How is it possible to argue that these benefits are related to mulberry leaves and not to the diet, since neither the calories consumed nor the composition of the diet followed during the 12 weeks of treatment is known?

Likewise, how can it be argued that improved mitochondrial function is related to mulberry leaves and not a reduction in diet-induced lipotoxicity?

In my view, this is too big a limit to accept publication.

Does the research team have data showing that these effects are not due to a change in diet but to DNJ?

Minor Revision:

In your paper, you demonstrate an effect of the mulberry leaf on Nrf-2.

Given the importance of Nrf2 on redox homeostasis in a condition of increased oxidative stress such as diabetes, have you considered evaluating the impact of the DNJ molecule on factors related to oxidative stress?

Considering that, for example, in the literature, it is known that the activation of JNK caused by oxidative stress in the muscle can induce insulin resistance already in a condition of pre-diabetes.

Round 2

Reviewer 1 Report

The manuscript has been improved but it is unacceptable to write under section 3.4 that "The results will be added when we receive the report from Institute of Nutrition, Mahidol University." The antioxidative properties of the leaf extract do not add much to the overall results. Remove this section.

Author Response

Thank you very much for reviewing and considering our manuscript for the revision. We really appreciate your kind suggestions and comments. We apologize for the part of result under section 3.4 Antioxidative property of mulberry leaf extract which we added. We absolutely agree with you, and we have already removed section 3.4 on page 10 and partial of section 2.1 regarding the antioxidative effect on page 2 from this manuscript. The corrections were prepared and defined by the yellow highlight. All the reviewer comments are very important to improve this paper appropriate for publication in the “Nutrients”. Please kindly again recommend any other additional points for the publication. Ultimately, we hope that our correction would be satisfied, and the paper will be published in the Nutrients soon.  

Reviewer 2 Report

Following the authors' responses to the requested reviews, I believe the manuscript is acceptable.

I would like to thank the authors for the comprehensive answers and clarifications provided.